# Neuroprotection of NRF2 against Ferroptosis after Traumatic Brain Injury in Mice

**DOI:** 10.3390/antiox12030731

**Published:** 2023-03-16

**Authors:** Hao Cheng, Pengfei Wang, Ning Wang, Wenwen Dong, Ziyuan Chen, Mingzhe Wu, Ziwei Wang, Ziqi Yu, Dawei Guan, Linlin Wang, Rui Zhao

**Affiliations:** 1Department of Forensic Pathology, China Medical University School of Forensic Medicine, Shenyang 110122, China; 2Liaoning Province Key Laboratory of Forensic Bio-Evidence Sciences, Shenyang 110122, China; 3Collaborative Laboratory of Intelligentized Forensic Science, Shenyang 110034, China; 4School of Forensic Medicine, Kunming Medical University, Kunming 650500, China

**Keywords:** NRF2, ferroptosis, traumatic brain injury (TBI), free iron, antioxidant

## Abstract

Ferroptosis and iron-related redox imbalance aggravate traumatic brain injury (TBI) outcomes. NRF2 is the predominant transcription factor regulating oxidative stress and neuroinflammation in TBI, but its role in iron-induced post-TBI damage is unclear. We investigated ferroptotic neuronal damage in the injured cortex and observed neurological deficits post-TBI. These were ameliorated by the iron chelator deferoxamine (DFO) in wild-type mice. In *Nrf2*-knockout (*Nrf2^−/−^*) mice, more sever ferroptosis and neurological deficits were detected. Dimethyl fumarate (DMF)-mediated NRF2 activation alleviated neural dysfunction in TBI mice, partly due to TBI-induced ferroptosis mitigation. Additionally, FTH-FTL and FSP1 protein levels, associated with iron metabolism and the ferroptotic redox balance, were highly NRF2-dependent post-TBI. Thus, NRF2 is neuroprotective against TBI-induced ferroptosis through both the xCT-GPX4- and FTH-FTL-determined free iron level and the FSP1-regulated redox status. This yields insights into the neuroprotective role of NRF2 in TBI-induced neuronal damage and its potential use in TBI treatment.

## 1. Introduction

Traumatic brain injury (TBI) is caused by external mechanical forces and has a high rate of disability and mortality. More than 10 million people worldwide suffer from TBI every year [1,2]. Since many factors affect the consequences of brain injury, TBI pathogenesis requires full elucidation. Many types of cell death are involved in neuronal damage after TBI, including necrosis, apoptosis, pyroptosis, and necroptosis [3,4,5,6]. It has been proven that TBI-induced neuronal necrosis and parenchymal hemorrhage release a large amount of free iron into the peripheral space, which further leads to ferroptosis due to excessive oxidative damage to neurons and glial cells [7,8]. Increased levels of antioxidant factors exert protective roles against ferroptosis [9,10], further indicating that ferroptosis is closely related to the disturbance of cellular redox homeostasis. Recent studies, including our own, have indicated that ferroptosis plays an important role in neuronal death and neural dysfunction after TBI [11,12,13], intracerebral hemorrhage [14], and spinal cord injury [15].

Iron is important in the biological process of ferroptosis. Excess free iron induces lipid peroxidation by promoting free radical production and impairing biomembrane function, leading to GSH depletion, increased COX2 levels, and toxic lipid peroxidation products, such as 4-HNE and MDA [16,17,18,19], which in turn lead to tissue edema, cytotoxic oxidative damage, and even cell death in damaged sections [20,21,22,23]. Ferroptosis, a newly recognized regulatory form of cell death associated with lipid peroxidation and iron metabolism disorders, is attributed to lethal lipid reactive oxygen species (ROS) regulated by glutathione peroxidase 4 (GPX4), and is inhibited by iron chelators and lipophilic antioxidants [16,24,25,26]. Under physical conditions, iron is stably stored in cells in the form of ferritin (consisting of FTH, which acts as a ferroreductase, and FTL, which stores large amounts of iron) [27]. Ferritin is the center of iron metabolism. It has been reported that the loss of FTH results in oxidative stress and impairs cortical iron homeostasis in mice [28,29]. Mutations in FTL contribute to brain iron dysregulation, early morphological signs of neurodegeneration, and motor coordination deficits [30]. Therefore, ferritin plays an important role in the stability of iron levels and in neuroprotection.

Nuclear factor erythroid-derived 2-related factor 2 (NRF2, NFE2L2) is a dominant member of the CNC-bZIP family and a key transcription factor that regulates the balance of oxidative stress in the body [31]. Many studies have shown that NRF2 exerts neuroprotective effects by antagonizing ferroptosis in rodents with spinal cord injury, subarachnoid hemorrhage [32,33], and TBI [34]. *Nrf2* deficiency in mice leads to the aggravation of redox imbalance [34,35], while NRF2 agonists significantly ameliorate TBI-induced mitochondrial dysfunction, oxidative damage, and inflammatory response [36,37,38]. Recent studies have provided evidence that NRF2 inhibits intracellular iron accumulation [39], lipid peroxidation, and ferroptosis in neural systems in both in vitro [40] and in vivo models [33,41].

Although previous studies have linked NRF2 with iron-related pathological progression, the exact role of NRF2 after TBI remains unclear. To illustrate the potential role of NRF2 in iron-related damage after TBI, wild-type (WT), *Nrf2*-knockout (*Nrf2^−/−^*) and dimethyl fumarate (DMF) [42]-treated mice were used to establish a controlled cortical impact (CCI) model and, subsequently, to detect iron metabolism and neuronal ferroptosis after TBI. Our study aimed to provide new evidence regarding the neuroprotective mechanism of NRF2. We showed that NRF2 is neuroprotective against TBI-induced ferroptosis through both the FTH-FTL-determined free iron level and the xCT-GPX4- and FSP1-determined redox statuses. Thus, our results shed new light on strategies for the treatment of TBI by targeting NRF2.

## 2. Materials and Methods

### 2.1. Animals

Adult male C57BL/6J mice were obtained from China Medical University, and *Nrf2*-knockout (*Nrf2^−/−^*) mice (20–25 g) were gifted by Dr. Jingbo Pi (School of Public Health, China Medical University). Mice were housed in a pathogen-free room, which was maintained at a controlled temperature (23 ± 1 °C) on a 12 h light/dark cycle, with water and feed available freely. Genotyping was performed by polymerase chain reaction (PCR) using genomic DNA isolated from tail clips, as previously described [43] (Appendix A). All procedures were performed in accordance with the National Guidelines for the Care and Use of Laboratory Animals. Animal experiments were reviewed and approved by the China Medical University Animal Care and Use Committee (AUP: KT2020135).

### 2.2. Model Handling and Sample Collection

We used 72 wild-type (WT) and 54 *Nrf2^−/−^* mice (8–12 weeks old) in this study. The mice were randomly divided into the following groups: WT (*n* = 18), WT+Vehicle (saline or methylcellulose) (*n* = 36), WT+DFO (deferoxamine dissolved in saline) (*n* = 18), WT+DMF (dimethyl fumarate dissolved in methylcellulose) (*n* = 18), *Nrf2^−/−^
*(*n* = 18), *Nrf2^−/−^*+Vehicle (*n* = 18), and *Nrf2^−/−^*+DFO (*n* = 18). In addition, each group of mice was subdivided into sham and TBI (1, 3 days post-injury [dpi]) subgroups (*n* = 6/subgroup). In the TBI group, mice were placed in the prone position on the stereotaxic apparatus after being anesthetized with 4% isoflurane in oxygen. A craniotomy was induced midway between the bregma and λ lateral to the left sagittal suture, and a vertical impact on the cortex was made using a controlled cortex impact (CCI) apparatus (PinPoint™ PCI3000, Hatteras Instruments, Grantsboro, NC, USA) with the following impact parameters: diameter impactor, 3 mm; depth, 1 mm; velocity, 1.5 m/s; and residence time, 50 ms. In addition, mice in the sham group underwent the same operation, but without cortical impact. The administration of DFO [44] (50 mg/kg/day, intraperitoneal, MedChemExpress, Shanghai, China) and DMF [45] (50 mg/kg/day, intragastric, MedChemExpress, Shanghai, China) was based on a previous study. A schematic flow diagram of the grouping and treatment is shown in Appendix A. The mice were then anesthetized with pentobarbital sodium and sacrificed at the indicated times. Ipsilateral cortex tissues were collected after heart perfusion with cold phosphate-buffered saline and stored at −80 °C for protein and gene analysis. For morphological analysis, mice were perfused with 4% paraformaldehyde. Staining images were captured using a Zeiss Axio Scan.Z1 confocal microscope system (Zeiss, Jena, Germany).

### 2.3. Immunofluorescence, TUNEL, and Fluoro-Jade C Staining

Immunofluorescence (IF) staining was performed as described previously [46]. To detect damage to cortical neurons, co-staining with NeuN and TUNEL was performed using a TUNEL BrightGreen Apoptosis Detection Kit (A111-01; Vazymem, Nanjing, China). The primary and secondary antibodies which we used are listed in Appendix A. Fluoro-Jade C (FJC) staining was conducted to detect neuronal degeneration in the injured cortical tissues according to the manufacturer’s instructions (#AG310, Millipore, Burlington, MA, USA).

### 2.4. Nissl Staining and Perl’s Staining

Nissl staining was performed according to the manufacturer’s instructions (Beyotime Biotechnology, Shanghai, China). Perl’s staining was conducted to detect iron in neurons according to the manufacturer’s instructions (G1422; Solarbio, Beijing, China), and the staining signal was developed using 3,3-diaminobenzidine. Stained images were acquired using a Zeiss Axio Scan.Z1 confocal microscope system (Zeiss, Jena, Germany).

### 2.5. Iron Content Determination

The iron level in the ipsilateral cortex was detected according to the instructions for the reagent (ab83366, Abcam, Cambridge, UK), protein concentration was determined using a BCA assay kit, and the amount of iron was normalized to the total protein level and expressed as iron level (nmol)/total protein level (mg).

### 2.6. Western Blotting Analysis

Ipsilateral injured cortices was collected and lysed in RIPA buffer (Beyotime, Shanghai, China) with protease inhibitor (PMSF) (Beyotime, Shanghai, China), and protein concentration was determined using the BCA assay kit (Beyotime, Shanghai, China). Western blotting was performed as previously described [46]. The relative band intensity was quantified using NIH ImageJ software and normalized to β-ACTIN. The antibodies which we used are listed in Appendix A.

### 2.7. Glutathione Detection

Glutathione (GSH) content in the ipsilateral injured cortex was detected using GSH assay kit (A006-1-1; NanJing JianCheng Bioengineering Institute, Nanjing, China) as previous described [29]. The protein concentration of the ipsilateral injured cortex was determined by a BCA assay kit (Beyotime, Shanghai, China), and GSH content was expressed as GSH level (μmol)/total protein (mg).

### 2.8. Transmission Electron Microscopy

The mitochondrial ultrastructure of the neurons in the injured cortex was examined by transmission electron microscopy (TEM), as described previously [15]. Ultrathin sections of tissues were prepared and visualized using a transmission electron microscope (Philips CM120, Holland).

### 2.9. Quantitative Real-Time PCR

The total RNA of the injured cortices was isolated with TRIzol reagent (279510, Thermo Fisher Scientific, CA, USA) and quantitatively determined using NanoDrop 2000C (Thermo Fisher Scientific, USA). Reverse transcription to cDNA was performed using the PrimeScript™ RT reagent kit (TRR047A, Takara Biotechnology, Japan), and RT-qPCR was performed using the SYBR^®^ Premix Ex Taq™ II RT-PCR kit (RR820A, Takara Biotechnology, Japan) for quantity analysis of the mRNA. mRNA levels were normalized to *β-Actin*. The primer sequences which we used are listed in Appendix A.

### 2.10. Assessment of Neuronal Function

Neurological severity scores (NSS) were used to assess the motion function of mice after TBI as previously described [47]. We monitored mice for the presence of mono- or hemiparesis, inability to walk on a 3-centimeter-wide beam, inability to walk straight, and loss of startle behavior. Higher scores indicated more severe damage to the mice. In addition, the pole test was used to further assess the motion function of the mice after TBI. The time to climb the round rod was recorded; the longer the time, the more serious the injury to the mice [48].

### 2.11. Statistical Analysis

Data are presented as the mean ± SD. The number of positive cells was counted and analyzed independently by researchers who were not involved in the trials using ImageJ software (version 6.0; National Institutes of Health). Data between groups were compared using two-way analysis of variance followed by Tukey’s post hoc multiple comparison test. Data were compared between two groups using Student’s *t*-test. Statistical analyses were performed using GraphPad Prism 8.0 (GraphPad Software, La Jolla, CA, USA), with *p* < 0.05 considered statistically significant.

## 3. Results

### 3.1. Ferroptosis Is Related to Neuronal Damage and Dysfunction after TBI

TBI results in diffused necrosis and vascular rupture, which contribute to iron overload in the injured tissues. To confirm iron deposition in neurons after TBI in our CCI model, we performed co-staining with Nissl and iron and detected robust iron deposition in the ipsilateral cortical neurons of mice at 1 and 3 days post-injury (dpi) (Figure 1A).

To explore the neuronal toxicity of the accumulated iron, the iron chelator deferoxamine (DFO) was used to scavenge the free iron. The deposition of iron in the injured cortex declined sharply after treatment (Figure 1B). This was accompanied by a reduction in TUNEL-positive neurons in the peripheral lesion area (Figure 1C,D). These results demonstrated that TBI-induced neuronal damage is closely associated with the iron overload.

To confirm the occurrence of ferroptosis after TBI, we performed double-immunostaining for ferroptotic markers and NeuN. We found that 4-HNE and COX2 levels were sharply increased in NeuN (+) cells in the periphery of lesion sites after TBI, which was ameliorated by DFO treatment (Figure 2e and Appendix A, lower panels). In addition, the expression of the ferroptosis-related genes *Acsf2* and *Ptgs2* was upregulated after TBI; this was suppressed by DFO treatment (Figure 1F).

Furthermore, a neurological assessment was conducted to evaluate the influence of iron overload on neural function in the mice, as shown in Appendix A. DFO treatment alleviated both motor dysfunctions (Appendix A) after TBI in mice. With these results taken together, we confirmed that ferroptosis contributes to neuronal damage and neurological dysfunction following TBI.

### 3.2. Nrf2 Deficiency Aggravates Neurological Dysfunction and Neuronal Damage after TBI in Mice

NRF2 has been shown to be a key neuroprotective factor against TBI in mice [49,50]. Robust expression of NRF2 occurs in neurons at 3 dpi in mice [51]. To confirm the neuroprotective roles of NRF2, WT and *Nrf2^−/−^* mice were used to examine neurological function after TBI. The neurological severity score (NSS) and the pole test were applied to evaluate the motor function of the mice after 3 dpi, and the results showed that the NSS of the *Nrf2^−/−^* mice was higher than that of the WT mice (Figure 2a), as was the average time for climbing rods (Figure 2b), suggesting that *Nrf2* deletion aggravated TBI-induced motor dysfunction in mice. Subsequently, the spatial memory of TBI mice at 14 dpi was evaluated using the Barnes maze, and the movement traces of the mice on the platform were recorded (Figure 2c). The escape latency, travel distance, and number of errors were significantly different between *Nrf2^−/−^* and WT mice after TBI (Figure 2d), suggesting that *Nrf2* deficiency aggravated neurological dysfunction and spatial memory impairment caused by TBI. Thereafter, fluoro-jade C (FJC) staining was used to reveal neuronal damage in the ipsilateral cortex, as shown in Figure 2c,d. *Nrf2* knockout exacerbated TBI-induced neuronal degeneration after TBI. In addition, evaluation of the co-localization of TUNEL and NeuN in the injured cortex demonstrated an increase in TUNEL-NeuN double-positive cells after TBI (Figure 3a,b), which was aggravated by *Nrf2*-knockout, accounting for the aggravated neurological damage after TBI in *Nrf2*-knockout mice. Taken together, *Nrf2* deletion aggravates neurological deficits and neuronal damage in TBI mice.

### 3.3. NRF2 Deletion Aggravates TBI-Induced Ferroptosis

NRF2 regulates oxidative stress and neuroinflammation, and plays a neuroprotective role after TBI [50]. To determine whether NRF2 is associated with the pathogenesis of iron accumulation and ferroptosis after TBI, we first detected iron deposition and ferroptosis in *Nrf2^−/−^* mice at 3 dpi. We examined iron levels in the ipsilateral injured cortex. Compared to *WT* mice, *Nrf2*-deficient mice showed higher levels of iron at 3 dpi (Figure 3a). Furthermore, increased levels of 4-HNEand COX2 (Figure 3b,c) and decreased levels of GSH (Figure 3d) were observed in the injured cortices of *Nrf2^−/−^* mice as compared with WT mice. The mRNA levels of ferroptosis-related genes, such as *Acsf2* and *Ptgs2*, were further increased by TBI in *Nrf2*-knockout mice (Figure 3e). In addition, transmission electron microscopy (TEM) was used to explore the ultrastructure of the injured neurons at 3 dpi. As shown in Figure 3f,g, exacerbated shrinkage, cristae disappearance, and high electron density of mitochondria in neurons were observed after TBI, and these effects were aggravated in *Nrf2^−/−^* mice. These results suggested that *Nrf2*-deletion enhanced TBI-induced iron accumulation and exacerbated neuronal lipid peroxidation and ferroptosis after TBI.

### 3.4. DMF Mitigates TBI-Induced Ferroptosis and Neurological Deficits

To further clarify the protective role of NRF2 in TBI-induced ferroptosis, we treated mice with the NRF2 agonist dimethyl fumarate (DMF). DMF treatment increased the levels of NRF2 and the expression of *Nrf2*; it also regulated downstream genes, such as *Ho-1*, *Nqo1*, *Gclc*, and *Gclm* (Appendix A). Moreover, oxidative products or protein levels of 4-HNE and COX2 in the injured cortices were decreased after the use of DMF (Figure 4a,b). In line with this, the immunoreactivity of 4-HNEand COX2 in neurons was also downregulated (Figure 4c and Appendix A). In contrast, the level of GSH increased at 3 dpi after DMF treatment (Figure 4d). Consequently, a reduction in damaged neurons (TUNEL-positive) was observed after treatment with DMF (Appendix A). The mRNA levels of *Acsf2* and *Ptgs2* were decreased in the injured cortices after DMF administration (Figure 4e). Our data indicate that DMF alleviated TBI-induced lipid peroxidation and ferroptosis after TBI.

In addition, we evaluated the effects of DMF on neurological function after TBI. As shown in Figure 4f,g, DMF improved the NSS and pole-climbing ability of TBI mice. Our results demonstrated that NRF2 activation improved neurological function, at least in part, by alleviating TBI-induced ferroptosis in mice.

### 3.5. NRF2 Regulates Ferroptosis through FTH-FTL, xCT-GPX4, and FSP1 after TBI

To explore the possible molecular mechanism by which NRF2 exerts effects in TBI-induced ferroptosis, we examined the impact of some candidate factors on the pathways of iron metabolism at 3 dpi. We found that NRF2 deficiency resulted in decreased FTH and FTL, whereas both the protein (Figure 5a,b) and mRNA levels (Figure 5c,d) were increased by DMF.

Since xCT and GPX4 are recognized as important factors for the synthesis and function of glutathione [52], the protein and mRNA levels of xCT and GPX4 were further investigated. As shown in Figure 4d and Figure 5a,b,e,f, a decrease in xCT, GPX4, and GSH caused by TBI was reversed after the administration of DMF, whereas the levels of xCT and GPX4 were decreased in *Nrf2*-knockout mice, particularly after TBI, demonstrating that xCT and GPX4 levels are highly dependent on NRF2 (Figure 4d and Figure 5a,b,e,f).

Because ferroptosis suppressor protein 1 (FSP1) has been recognized as an important ferroptotic inhibitor [53,54], we tested the expression of FSP1 at the protein and mRNA levels. As shown in Figure 5a,b,g, FSP1 was positively regulated by NRF2 (Figure 5a,b,g). Thus, NRF2 affects iron metabolism and ferroptosis after TBI by regulating FTH-FTL, xCT-GPX4, and FSP1 levels.

## 4. Discussion

Iron-related redox imbalances and ferroptosis are important factors that aggravate TBI outcomes. NRF2 has been demonstrated to be the predominant transcription factor regulating oxidative stress and inflammation after TBI [49,50]. However, the roles and mechanisms of NRF2 in iron-dependent regulated cell death after TBI have been unclear. In this study, we demonstrated a novel neuroprotective effect of NRF2 involving inhibition of iron overload and antagonization of ferroptosis after TBI, which was partially mediated by FTH-FTL and xCT-GPX4 FSP1 (Figure 6).

Ferroptosis is a newly recognized form of regulated cell death that is mainly characterized by the availability of redox-active iron and the loss of lipid peroxidative repair capacity [55]. We and others have found that ferroptosis contributes to neuronal loss and neurological dysfunction after TBI [11,13]. TBI leads to the accumulation of free iron in the interstitial space due to hemorrhage and cellular necrosis. Excess iron has been proven to be involved in the production of ROS, lipid peroxidation, inflammation, and autophagy in TBI models [23,56]. Iron accumulation is associated with deficits in spatial learning, spatial memory, and long-term prognosis, which are ameliorated by the use of ferroptotic inhibitors in both TBI [57] and craniocerebral injury [58]. In our present study, TBI-induced iron accumulation and lipid peroxidation led to neuronal impairment and abnormal neurological function. These results indicate that iron overload and ferroptosis are closely associated with TBI-induced neurological dysfunction.

NRF2 is the most important nuclear transcription factor involved in redox balance regulation. Loss of NRF2 causes sensitivity to oxidative damage [59]. Many studies have shown that NRF2 has a wide range of protective effects against cortical lesions, edema, and motor and neurological damage after TBI [60,61], which are related to anti-oxidative, anti-inflammatory, anti-apoptotic, and ubiquitination regulatory activities [31,49,62]. Considering that lipid peroxidation is the main characteristic of ferroptosis, NRF2 would be involved in the pathogenesis of TBI-induced neural damage. Recent studies have provided evidence that NRF2 may participate in the regulation of ferroptosis [49]. In the present study, we found that NRF2 is not only involved in iron metabolism, but also in the antioxidant process of ferroptosis. The present study provides solid evidence for a novel neuroprotective role of NRF2.

Excess iron reacts with hydrogen peroxide to generate vast numbers of hydroxyl radicals, leading to tissue impairment [63]. Ferritin, composed of FTH and FTL, which that can accommodate up to 4500 iron atoms [64], is a ubiquitous intracellular spherical iron storage protein that plays a key role as a dynamic iron buffer in organisms. By isolating and stabilizing excess free iron, ferritin exerts an antioxidative effect and reduces the damage caused by iron overload [65,66]. Many studies have shown that NRF2 plays a significant role in iron storage and iron metabolism [50,67], and this is supported by the evidence that NRF2 acts as a transcription factor to regulate *Fth* and *Ftl* in rat livers [68]. Moreover, the presence of antioxidant response elements (AREs) has been confirmed in the promoter regions of mouse *Ftl* and *Fth* [69,70]. In this study, the loss of *Nrf2* resulted in a decrease in FTH and FTL levels after TBI, followed by an increase in free iron levels, which might have contributed to neuronal dysfunction. In this work, we use deferoxamine (DFO) as an iron chelator after TBI, although there are queries regarding whether DFO can penetrate BBB, alleviate TBI-induced ferroptosis, and remove excessive iron, as our previous and present studies have shown in WT mice [13]. Thus, we believe that at least post-TBI BBB damage leads to easier entry for DFO, which may be beneficial to its anti-ferroptotic and neuroprotective effects.

Recent studies have provided evidence that FSP1 plays an antioxidant role parallel to the xCT-GPX4 pathway, and that it exerts an inhibitory effect on lipid peroxidation and ferroptosis [71,72]. The Xc-antioxidant regulatory system (cystine–glutamate antiporter system) plays a significant upstream role in the ferroptosis signaling cascade [73,74,75]. It consists of light chain xCT (SLC7A11) and heavy chain 4F2 (SLC3A2), which transport extracellular cysteine into cell plasma for glutathione biosynthesis [73,74,75]. 

Previous studies have found that NRF2 upregulates xCT (SLC7A11) [76,77] and exerts neuroprotective effects by inhibiting ferroptosis [76,78]. GPX4, a downstream factor of xCT, plays an irreplaceable role in ferroptosis pathogenesis. Loss of GPX4 is a crucial event in ferroptosis [79,80], and promotes cognitive impairment [81]. 

Considering that both GPX4 and GCLC/GCLM (key enzymes for synthesizing glutathione) are transcriptionally regulated by NRF2 [82,83], we confirmed that the expression of xCT and GPX4 was highly dependent on the activation of NRF2 by using *Nrf2*-knockout and DMF-treated mice. This was consistent with a previous study showing that activation of NRF2 promoted functional recovery by increasing GPX4 levels after spinal cord injury [33]. Additionally, FSP1, a newly discovered ferroptotic antagonist, plays an antioxidant and anti-ferroptosis role by reducing ubiquitin (CoQ) to antioxidative dihydroubiquitin (CoQH2) [71,72]. Herein, the expression of FSP1 was interrupted by *Nrf2*-knockout and upregulated by the administration of DMF, which was consistent with the previous identification of *Fsp1* as a downstream gene of NRF2 [84]. Therefore, we affirmed that NRF2 exerts neuroprotective effects against TBI-induced ferroptosis by regulating not only the xCT-GPX4 pathway, but also the GPX4-independent FSP1.

In the present study, we expanded the neuroprotective effects of NRF2 after TBI on the regulation of iron metabolism and ferroptosis. However, this study had some limitations. Although the protective effects and the potential mechanisms of NRF2 on TBI- induced neuronal ferroptosis were discussed in this study, there remains a need to explore the potential roles of NRF2 in iron metabolism and ferroptosis in glial cells. In addition, because only global *Nrf2*-knockout and DMF-treated mice were used, the influence of *Nrf2*-deleted glia on iron metabolism and ferroptosis of injured neurons cannot be neglected.

## 5. Conclusions

Neuronal ferroptosis contributes to neurological dysfunction after TBI in mice. NRF2 has protective effects on iron deposition and neuronal ferroptosis after TBI. These effects are exerted by reducing iron metabolism through FTH-FTL and inhibiting lipid peroxidation through FSP1. Our research provides evidence for the novel neuroprotective roles of NRF2 and sheds new light on strategies for the treatment of TBI by targeting NRF2.

## Figures and Tables

**Figure 1 antioxidants-12-00731-f001:**
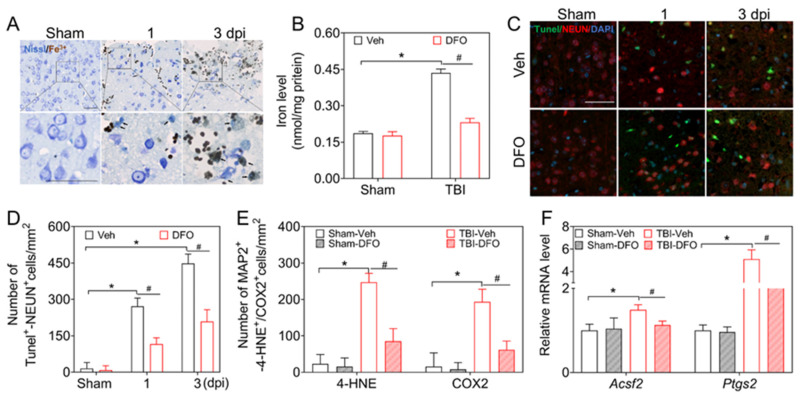
Deferoxamine (DFO) improves ferroptosis after traumatic brain injury (TBI). (**A**) Representative images of iron staining in Nissl-positive cells at 1 and 3 dpi. *n* = 3, bar = 50 μm. (**B**) Iron levels in the injured cortex of WT mice at 3 dpi with Veh or DFO treatment, *n* = 6. (**C**) Co-localization of NEUN and TUNEL in injured cortex at 1 and 3 dpi after Veh or DFO treatment. *n* = 3, bar = 50 μm. (**D**) Diagram of TUNEL-NeuN-positive cells in the injured cortex treated with DFO and Veh at 1 and 3 dpi, *n* = 3. (**E**) Diagram of the 4-HNE- and COX2-positive cells in injured cortical neurons at 3 dpi following Veh or DFO treatment, *n* = 3. (**F**) mRNA levels of ferroptosis-related genes *Acsf2* and *Ptgs2* in the injured cortex of mice treated with Veh or DFO, *n* = 6. Data are expressed as mean ± SD, * *p* < 0.05 vs. sham—Veh, ^#^ *p* < 0.05 vs. TBI—Veh. Two-way analysis of variance was followed by Tukey’s post hoc multiple comparison test for comparisons between more than two groups. Veh, vehicle; DFO, deferoxamine.

**Figure 2 antioxidants-12-00731-f002:**
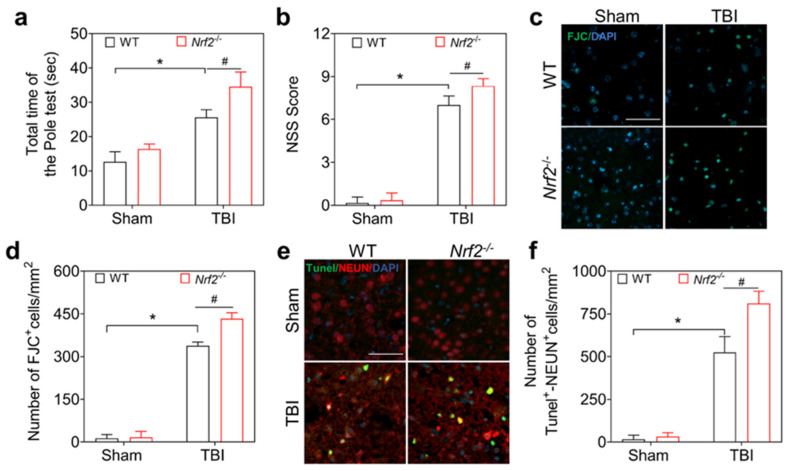
*Nrf2* deficiency aggravates motor function and neuronal damage after TBI in mice. (**a**) Time spent in the pole test in WT and *Nrf2^−/−^* mice at 3 dpi, *n* = 6. (**b**) NSS of WT and *Nrf2^−/−^* mice at 3 dpi, *n* = 6. (**c**) Representative image of fluor-jade C (FJC) staining in the cortical lesions of WT and *Nrf2^−/−^* mice at 3 dpi. *n* = 3, Bar = 50 μm. (**d**) Quantitative analysis of number of FJC^+^ cells, *n* = 3. (**e**) Co-localization of TUNEL and NeuN in injured cortices of WT and *Nrf2^−/−^* mice at 3 dpi. Bar = 50 μm. (**f**) Quantitative analysis of the number of TUNEL-NeuN-positive cells in the injured cortex of WT and *Nrf2^−/−^* mice at 3 dpi, *n* = 3. Data are expressed as mean ± SD, * *p* < 0.05 vs. WT—sham, ^#^ *p* < 0.05 vs. WT—TBI. Two-way analysis of variance followed by Tukey’s post hoc multiple comparison test for comparisons between more than two groups. WT, wild-type; TBI, traumatic brain injury.

**Figure 3 antioxidants-12-00731-f003:**
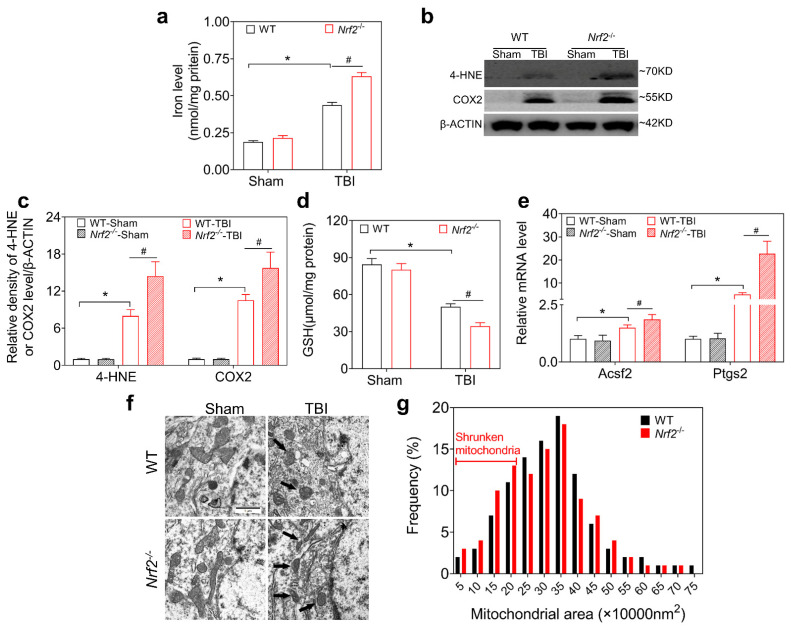
*Nrf2*-deletion aggravates iron overload, lipid peroxidation, and cell damage after TBI. (**a**) Iron levels in the injured cortices of WT and *Nrf2^−/−^* mice at 3 dpi, *n* = 6. (**b**) Representative band of 4-HNE and COX2 levels in the injured cortices of WT and *Nrf2^−/−^* mice at 3 dpi. (**c**) Relative intensities of 4-HNE and COX2 in the injured cortices of WT and *Nrf2^−/−^* mice at 3 dpi., *n* = 3. (**d**) GSH levels in the injured cortices of WT and *Nrf2^−/−^* mice at 3 dpi, *n* = 6. (**e**) mRNA levels of the ferroptosis-related genes *Ptgs2* and *Acsf2* in the injured cortices of WT and *Nrf2^−/−^* mice at 3 dpi, normalized to *β-Actin*, *n* = 6. (**f**) Representative ultrastructure of mitochondria in neurons examined by transmission electron microscopy after TBI at 3 dpi. *n* = 3, bar = 1 μm. (**g**) Frequency of shrunken mitochondria in neurons after TBI in WT and *Nrf2^−/−^* mice. More shrunken mitochondria in the 5–20 × 10^4^ nm^2^ group were observed in *Nrf2^−/−^* mice than in the WT mice. Data are expressed as mean ± SD, * *p* < 0.05 vs. WT—sham, ^#^ *p* < 0.05 vs. WT—TBI. Two-way analysis of variance was followed by Tukey’s post hoc multiple comparison test for comparisons more between than two groups. WT, wild-type; TBI, traumatic brain injury; SD, standard deviation.

**Figure 4 antioxidants-12-00731-f004:**
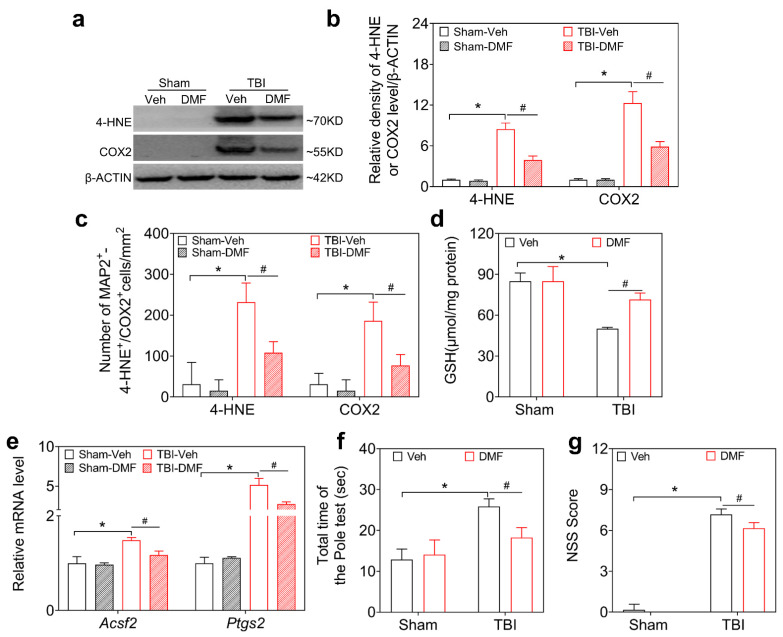
DMF improves ferroptosis and neurological dysfunction after TBI. (**a**) Representative band of 4-HNE and COX2 levels in the injured cortices of WT mice treated with Veh or DMF at 3 dpi. (**b**) Relative intensities of 4-HNE and COX2 in the injured cortices of WT mice treated with Veh or DMF at 3 dpi, *n* = 3. (**c**) Number of 4-HNE- or COX2-MAP2 cells in the injured cortices of WT mice treated with Veh or DMF at 3 dpi, *n* = 3. (**d**) GSH level in the injured cortices of WT mice treated with Veh or DMF at 3 dpi, *n* = 6. (**e**) mRNA level of ferroptosis-related genes *Acsf2* and *Ptgs2* in the injured cortices of WT mice treated with Veh or DMF at 3 dpi, normalized to *β-Actin*, *n* = 6. (**f**) Time spent in the pole test at 3 dpi in DMF-treated mice, *n* = 6. (**g**) NSS of mice at 3 dpi, *n* = 6. Data are expressed as mean ± SD, * *p* < 0.05 vs. Sham—Veh, ^#^ *p* < 0.05 vs. TBI—Veh. Two-way analysis of variance followed by Tukey’s post hoc multiple comparison test for comparisons including more than two groups. WT, wild-type; TBI, traumatic brain injury; Veh, vehicle; DMF, dimethyl fumarate; SD, standard deviation.

**Figure 5 antioxidants-12-00731-f005:**
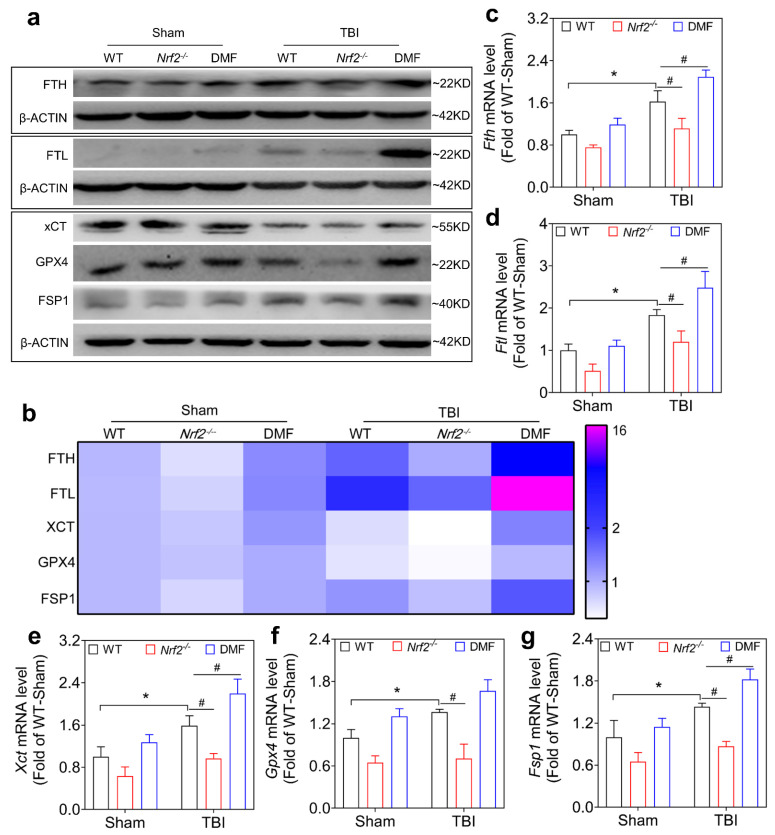
NRF2 regulates the expression of ferroptosis pathway-related FTH, FTL, xCT, GPX4, and FSP1. (**a**) Representative protein levels of FTH, FTL, xCT, GPX4, and FSP1 in the injured cortices of WT, *Nrf2^−/−^*, and DMF-treated mice at 3 dpi. (**b**) Heat map of relative intensity of FTH, FTL, xCT, GPX4, and FSP1, analyzed by GraphPad Prism 8.0 and normalized to β-ACTIN, *n* = 3. (**c**–**g**) mRNA levels of ferroptosis pathway-related genes Fth, Ftl, Xct, Gpx4, and Fsp1 in the injured cortices of WT, *Nrf2^−/−^*, and DMF-treated mice at 3 dpi, normalized to β-Actin, *n* = 6. Data are expressed as mean ± SD, * *p* < 0.05 vs. WT—sham, # *p* < 0.05 vs. WT—TBI. Two-way analysis of variance was followed by Tukey’s post hoc multiple comparison test for comparisons between more than two groups.

**Figure 6 antioxidants-12-00731-f006:**
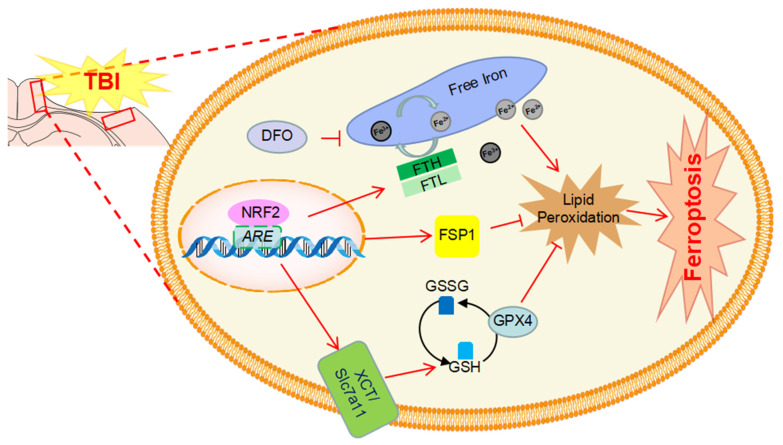
Schematic regulation diagram of NRF2 on ferroptosis after TBI. NRF2 regulates ferroptosis after TBI by modulating iron levels through FTH-FTL and redox status through xCT-GPX4, FSP1.

## Data Availability

All data are contained within the article or in the Appendix A. And data that relate to the present study are available from the corresponding author, Rui Zhao, upon reasonable request.

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
