# Peer review of "Neuroprotection of NRF2 against Ferroptosis after Traumatic Brain Injury in Mice"

_antioxidants, 2023, doi:10.3390/antiox12030731_

Round 1

Reviewer 1 Report

In previous work, the authors showed that ferroptosis plays a significant role in aggravating traumatic brain injury (TBI) outcomes. In the present work, they extend the study to analyse the role of NRF2. It confirms that ferroptosis occurs after TBI and it is alleviated by iron chelation. The effect is exacerbated in NRF2-KO mice and is partially alleviated by treatment with dimethyl fumarate, an agent that stimulates NRF2 expression. The work is potentially interesting, but it can be evaluated only after some major problems of the presentation are fixed.

- Material and methods should be after the introduction or after the discussion, not at the beginning of the work. Also, the template at the end of the manuscript is not necessary.

- Fig. 4, which should present data on the DFO treatment of NRF2-KO mice, is missing.

- Line 230: «…. Barnes maze, and the movement traces of the mice in the platform were recorded (Fig. 2c).» Fig 2c does not show the traces but FJC cell stain.

- Line 294: «protein levels of 4-HNE ...» 4-HNE is not a protein but an oxidation product of lipids. Correct throughout text and figures.

- Line 332: «As shown in Fig. 6a, b and Fig. 6e, f, a decrease in xCT and GPX4 ….» unfortunately data of xCT nad GPX4 are not in the figure.

- Methods. The mice used were all males? as reported in ref 19.

- DFO is not expected to cross BBB and reach the brain under normal conditions. In fact, patients on iron chelator can take DFO for years without any effect on brain iron. Thus the finding that DFO reduces brain iron in mice is surprising and must be commented on.  Maybe the authors can verify whether TBI causes BBB damage that would allow the chelator to enter the brain. Alternatively, they should use the deferiprone iron chelator known to cross BBB.

Author Response

Dear reviewer,

thank you for your valuable review for the manuscript, all of your comments have been response followed carefully in a PDF.Please see the attachment.

Reviewer 2 Report

This article presents a highly logical and promising study of the involvement of ferroptosis, proteins associated with iron metabolism and lipid peroxidation in TBI-induced neuronal degeneration on the one hand, and the neuroprotective potential of the NRF2 factor and the corresponding pathway on the other hand. The data are presented correctly and are described quite fully, the need for future additional studies of disorders in glial cells is noted. In my opinion, the article is almost ready for publication - I managed to find only a few minor errors that should be corrected:

1. Figure 2. Nrf2-deficiency aggravates motor functionand functions and neurons damage after TBI in mice

2. In capture to Supplementary Figure S2 - Representative double immunostaining images of MAP2 with 4-HNE (a), COX2 (a)?

3. Supplementary Table 1 – dilusion dilution

Author Response

(The authors gave the same response as above.)

Round 2

Reviewer 1 Report

Most of the suggested modifications have been applied and the manuscript can now be read.  A few minor points are to be addressed.

- correct the sentence in Line 593: "Herein, we think that at least post-TBI BBB disruption leads to easier entry of DFO, which attribute to its effect of anti-ferroptotic and neuroprotective."  BBB is not disrupted but only damaged. Which attribute... is not clear. 

- explain what Fig. 5b represents.

- In all legends to the figure for each plot there is an indication of the meaning of  * and #.  Since they always mean p<0.05 this can be put at the end of the legends.

Author Response

Dear reviewer,

thank you for your careful review for the manuscript, all of your comments have been response followed carefully in a PDF.Please see the attachment.
